# EXPLORING TRANSFERABILITY OF PERTURBATIONS IN DEEP REINFORCEMENT LEARNING

## ABSTRACT

The use of Deep Neural Networks (DNN) as function approximators has led to striking progress for reinforcement learning algorithms and applications. At the same time, deep reinforcement learning agents have inherited the vulnerability of DNNs to imperceptible adversarial perturbations to their inputs. Prior work on adversarial perturbations for deep reinforcement learning has generally relied on calculating an adversarial perturbation customized to each state visited by the agent. In this paper we propose a more realistic threat model in which the adversary computes the perturbation only once based on a single state. Furthermore, we show that to cause a deep reinforcement learning agent to fail it is enough to have only one adversarial offset vector in a black-box setting. We conduct experiments in various environments from the Atari baselines, and use our single-state adversaries to demonstrate the transferability of perturbations both between states of one MDP, and between entirely different MDPs. We believe our adversary framework reveals fundamental properties of the environments used in deep reinforcement learning training, and is a tangible step towards building robust and reliable deep reinforcement learning agents.

## 1 INTRODUCTION

Building on the success of DNNs for image classification, deep reinforcement learning has seen remarkable advances in various complex environments Mnih et al. (2015); Schulman et al. (2017); Lillicrap et al. (2015). Along with these successes come new challenges stemming from the lack of robustness of DNNs to small adversarial perturbations to inputs. This lack of robustness is especially critical for deep reinforcement learning, where the actions taken by the agent can have serious real-life consequences (Levin & Carrie (2018)).

The initial work on adversarial examples in deep learning by Szegedy et al. (2014) showed that imperceptible perturbations added to images could be used to cause a DNN image classifier to misclassify, and hypothesized that this was a result of the DNN's complexity and nonlinearities. Follow-up work by Goodfellow et al. (2015) proposed a more computationally efficient fast gradient sign method (FGSM) for computing adversarial examples, while simultaneously explaining the presence of adversarial examples as a result of DNN classifiers learning approximately linear functions. The authors additionally argue that the presence of a common high-dimensional linear decision boundary is an explanation for the transferability of adversarial examples between models trained on different minibatches of the same training set or with different architecture. The transferability of perturbations themselves, in contrast to transferability of complete adversarial examples, was investigated by Moosavi-Dezfooli et al. (2017). The authors showed how to compute one universal perturbation from a minibatch of images that could be used to fool a classifier by adding it to each image in the validation set. The authors argue the existence of universal perturbations is explained by correlations between the local decision boundaries near many different points in the dataset.

More recently, a new body of research on adversarial perturbations for deep reinforcement learning agents has developed, demonstrating that it is possible to make a high-performing agent fail by adding perturbations to the agent's perception of the current state Lin et al. (2017); Sun et al. (2020); Huang et al. (2017); Pattanaik et al. (2018); Mandlekar et al. (2017). A major drawback of this prior work is the threat model, in which the attacker computes perturbations and adds them to the agent's state in real-time before the state is perceived by the agent. Such a threat model is not practically

feasible considering the heavy duty optimization methods used by Lin et al. (2017); Sun et al. (2020). To address this we propose a new threat model where the attacker computes just one adversarial offset vector based on a single state. Not only is this a more practical threat model from a security perspective, but the success of attacks in this model can be used to give insight into the transferability of adversarial examples and the properties of functions learned by trained deep reinforcement learning agents.

We believe the adversarial perspective is an initial step towards understanding the loss landscape and geometry of the decision boundaries of the algorithms in use, assessing the generalization capabilities of trained agents, and building robust and reliable deep reinforcement learning agents. For these reasons in our paper we focus on the adversarial perspective and make the following contributions:

- We introduce a framework of six unique adversary types encapsulating our new threat model where the attacker is restricted to computing an adversarial perturbation based on a single state.

- We use our framework to investigate the transferability of adversarial perturbations, both between states of the same MDP and between MDPs.

- We show with experiments in the Atari baselines that single state attacks have significant impact on agent performance.

- In particular, we show that by using only one perturbation computed based on a random state of a random episode from a completely different environment, it is possible to have a dramatic impact on the deep reinforcement learning agent's performance. This transferability of adversarial perturbations between environments implies that there is a correlation between the features learned by agents trained in different environments. While these results demonstrate that deep reinforcement learning agents are indeed learning representations that generalize across environments, they also demonstrate a critical vulnerability from the security point of view that can be exploited without any knowledge of the training environment or neural architecture.

## 2 RELATED WORK AND BACKGORUND

### 2.1 DEEP REINFORCEMENT LEARNING

In this paper we examine discrete action space MDPs that are represented by a tuple: $M = (\mathbb{S}, \mathbb{A}, \mathbb{P}, r, \gamma, s_0)$ where $\mathbb{S}$ is a set of continous states, $\mathbb{A}$ is a set of discrete actions, $\mathbb{P} : \mathbb{S} \times \mathbb{A} \times \mathbb{S} \to \mathbb{R}$ is the transition probability, $r : \mathbb{S} \times \mathbb{A} \to \mathbb{R}$ is the reward function, $\gamma$ is the discount factor, and $s_0$ is the initial state distribution. The agent interacts with the environment by observing $s \in \mathbb{S}$ and taking actions $a \in \mathbb{A}$. The goal is to learn a policy $\pi_\theta : \mathbb{S} \times \mathbb{A} \to \mathbb{R}$ which takes an action $a$ in state $s$ that maximizes the cumulative discounted reward $\sum_{t=0}^{T-1} \gamma^t r(s_t, a_t)$. For an MDP $M$ and policy $\pi(s, a)$ we call a sequence of state, action, reward tuples, $(s_i, a_i, r_i)$, that occurs when utilizing $\pi(s, a)$ in $M$ an episode. We use $p_{M,\pi}$ to denote the probability distribution over the episodes generated by the randomness in $M$ and the policy $\pi$.

### 2.2 ADVERSARIAL PERTURBATION METHODS

In this work we create the adversarial perturbations via two different methods. The first is the Carlini & Wagner (2017) formulation. In the deep reinforcement learning setup this formulation is,

$$\min_{s_{\text{adv}} \in \mathbb{S}} c \cdot J(s_{\text{adv}}) + \|s_{\text{adv}} - s\|_2^2 \tag{1}$$

where $s$ is the unperturbed input, $s_{\text{adv}}$ is the adversarially perturbed input, and $J(s)$ is the augmented cost function used to train the network. The second method we use to produce the adversarial examples is the elastic net regularization (EAD) Chen et al. (2018) adversarial formulation,

$$\min_{s_{\text{adv}} \in \mathbb{S}} c \cdot J(s_{\text{adv}}) + \lambda_1 \|s_{\text{adv}} - s\|_1 + \lambda_2 \|s_{\text{adv}} - s\|_2^2 \tag{2}$$

# 3 EXPLORING TRANSFERABILITY

In this work we introduce the notion of single-state transferability in deep reinforcement learning, and investigate its effects in state-of-the-art deep reinforcement learning environment baselines. We explore single-state transferability in our proposed adversary framework, which consists of computing an adversarial perturbation from a random state, and then adding that particular adversarial perturbation to several other states, both within the same MDP and across different MDPs. We refer to the transferability of adversarial perturbations between states in the same MDP as *state transferability*. Similarly, we refer to transferability of perturbations between states in different MDPs as *environment transferability*. To be able to fully explore the different types of transferability we define a framework with several adversary variants in Section 3.2.

## 3.1 THREAT MODEL

In our work we propose a threat model that could plausibly be deployed in a physical deep reinforcement learning setup. Physical adversaries for image classification have been extensively studied. Kurakin et al. (2017) showed that it is possible to fool a DNN by printing out an adversarial picture and then presenting the print out as an input to the classifier. Evtimov et al. (2018) demonstrated that placing stickers on an object can also make DNN image classifiers fail. Athalye et al. (2017) showed that one can physically build an adversarial object with a 3D printer that can fool a DNN classifier. Recently, Li B. et al. (2019) proposed a realistic physical perturbation, a camera sticker, to fool DNN image classifiers.

Previous work on adversary models in deep reinforcement learning has focused on computing the adversarial examples customized to each state Huang et al. (2017); Mandlekar et al. (2017); Pattanaik et al. (2018); Lin et al. (2017); Pinto et al. (2017); Sun et al. (2020). In particular, the adversary model of Lin et al. (2017); Sun et al. (2020) requires real-time access to the agents perception system, while using the highly computationally demanding Carlini & Wagner (2017) adversarial formulation to compute perturbations. Such a model, where the adversarial perturbation for each input state is computed and added in real time, is firstly not computationally efficient, and secondly not a realistic adversarial scenario in many settings of interest. In our threat model we propose to only have one offset vector to add to every clean image observed by the deep reinforcement learning agent. We believe this scenario to be more physically viable, since this single adversarial perturbation could be a change to the lens of the camera of the deep reinforcement learning agent.

## 3.2 ADVERSARIAL FRAMEWORK

In the following section we define six unique adversary models in our framework to evaluate our proposed threat model. Given a policy $\pi(s, a)$ we define $\eta(s, \pi(s, a))$ to be a perturbation computed based only on the state $s$ and the policy $\pi(s, a)$ in state $s$. In all the following definitions we fix a bound $\kappa$ and set

$$\epsilon(\eta) = \frac{\kappa}{\|\eta\|_p}. \tag{3}$$

**Definition 1:** *Individual state adversary*, $\mathcal{A}^{\text{individual}}$, is the adversary that calculates the perturbation custom to each state.

$$s^* = s + \epsilon(\eta) \cdot \eta(s, \pi(s, a)) \tag{4}$$

**Definition 2:** *Initial state adversary*, $\mathcal{A}^{\text{initial}}$, is the adversary that calculates the perturbation based on the initial state of the agent in the given episode, and adds the initial state perturbation to the rest of the states visited.

$$s^* = s + \epsilon(\eta) \cdot \eta(s_0, \pi(s_0, a)) \tag{5}$$

**Definition 3:** *Episode independent initial state adversary*, $\mathcal{A}_{\text{e}}^{\text{initial}}$, is the adversary that calculates the perturbation based on the initial state of the agent in a random episode $e$, and adds the initial state perturbation to all states visited in episode $e'$. Given an episode $e \sim p_M$, the perturbed state is,

$$s^* = s + \epsilon(\eta) \cdot \eta(s_0(e), \pi(s_0(e), a)) \tag{6}$$

where $M$ is the MDP and $s_0(e)$ is the initial state of episode $e$.

**Definition 4:** *Episode independent random state adversary*, $\mathcal{A}_e^{\text{random}}$, is the adversary that calculates the perturbation based on a random state of the agent in a random episode $e$, and adds the random state perturbation to all the states visited in episode $e'$. Given an episode $e \sim p_M$, the perturbed state is,

$$s^* = s + \epsilon(\eta) \cdot \eta(s_{\text{random}}(e), \pi(s_{\text{random}}(e), a)) \tag{7}$$

where $M$ is the MDP and $s_{\text{random}}(e)$ is a random state of episode $e$.

**Definition 5:** *Environment independent initial state adversary*, $\mathcal{A}_M^{\text{initial}}$, is the adversary that calculates the perturbation based on the initial state of an agent in a random episode of the MDP $M$, and adds the initial state perturbation to all of the states visited in an episode of the MDP $M'$. Given an episode $e \sim p_M$, the perturbed state is,

$$s^* = s + \epsilon(\eta) \cdot \eta(s_0(e, M), \pi(s_0(e, M), a)) \tag{8}$$

where $s_0(e, M)$ is the initial state of episode $e$ of MDP $M$.

**Definition 6:** *Environment independent random state adversary*, $\mathcal{A}_M^{\text{random}}$, is the adversary that calculates the perturbation based on a random state of the agent in a random episode of the MDP $M$, and adds the random state perturbation to all the states visited in an episode of the MDP $M'$. Given an episode $e \sim p_M$, the perturbed state is

$$s^* = s + \epsilon(\eta) \cdot \eta(s_{\text{random}}(e, M), \pi(s_{\text{random}}(e, M), a)) \tag{9}$$

where $s_{\text{random}}(e, M)$ is a random state of episode $e$ of MDP $M$.

## 4 EXPERIMENTS

In our experiments agents are trained with DDQN Wang et al. (2016) in the Atari environment Bellemare et al. (2013) from OpenAI Brockman et al. (2016). We average over 10 episodes in our experiments. We define the impact of the adversary on the agent by normalizing the performance drop as follows

$$\text{Impact} = \frac{\text{Score}_{\text{max}} - \text{Score}_{\text{adv}}}{\text{Score}_{\text{max}} - \text{Score}_{\text{min}}}. \tag{10}$$

Here $\text{Score}_{\text{max}}$ is the score of the baseline trained agent following the learned policy without the presence of an adversary, $\text{Score}_{\text{adv}}$ is the score of the agent under the influence of the adversary, and $\text{Score}_{\text{min}}$ is the score the agent receives when choosing the action minimizing its $Q(s, a)$ function in every state. All scores are recorded at the end of an episode. We chose this normalization because we observed that the agent can still receive non-zero reward even when choosing the worst action in each state.

In our framework the adversaries $\mathcal{A}^{\text{initial}}$, $\mathcal{A}_e^{\text{initial}}$ and $\mathcal{A}_e^{\text{random}}$ all explore different aspects of state transferability. $\mathcal{A}_e^{\text{random}}$ is the most natural test for transferability between states of one MDP, while $\mathcal{A}^{\text{initial}}$ and $\mathcal{A}_e^{\text{initial}}$ represent more restricted adversaries that only have access to the initial state. Similarly, $\mathcal{A}_M^{\text{random}}$ is the most natural test for environment transferability, while $\mathcal{A}_M^{\text{initial}}$ is a more restricted version where the adversary does not have access to arbitrary states in the MDP $M$. Therefore, higher impact in $\mathcal{A}_e^{\text{random}}$ and $\mathcal{A}_M^{\text{random}}$ indicates higher state and environment transferability respectively.

Table 1 and Table 2 show impact values of six different adversary models from our adversarial framework utilizing the Carlini & Wagner (2017) formulation and EAD formualtion respectively

for various environments from the Atari baselines. These results show a striking level of both state transferability and environment transferability. One can observe that while the EAD formulation has significant state transferability for all environments, the Carlini & Wagner (2017) formulation only has significant state transferability for CrazyClimber and Pong. Additionally, the EAD formulation also has higher environment transferability in all environments except Pong and Roadrunner.

Table 1: Impacts of Carlini & Wagner (2017) for the six different adversary definitions within the proposed adversarisal framework.

| Environments | $\mathcal{A}^{individual}$ | $\mathcal{A}^{initial}$ | $\mathcal{A}_e^{initial}$ | $\mathcal{A}_e^{random}$ | $\mathcal{A}_M^{initial}$ | $\mathcal{A}_M^{random}$ |
|---|---|---|---|---|---|---|
| BankHeist | 0.694 | 0.232 | 0.069 | 0.118 | 0.086 | 0.022 |
| RoadRunner | 0.876 | 0.674 | 0.755 | 0.649 | 0.064 | 0.919 |
| JamesBond | 0.038 | 0.048 | 0.086 | 0.019 | 0.942 | 0.304 |
| CrazyClimber | 0.646 | 0.637 | 0.601 | 0.956 | 0.014 | 0.036 |
| TimePilot | 0.734 | 0.334 | 0.316 | 0.228 | 0.068 | 0.017 |
| Pong | 0.998 | 1.0 | 1.0 | 1.0 | 0.007 | 0.966 |

Table 2: Impacts of EAD for the six different adversary definitions within the proposed adversarisal framework.

| Environments | $\mathcal{A}^{individual}$ | $\mathcal{A}^{initial}$ | $\mathcal{A}_e^{initial}$ | $\mathcal{A}_e^{random}$ | $\mathcal{A}_M^{initial}$ | $\mathcal{A}_M^{random}$ |
|---|---|---|---|---|---|---|
| BankHeist | 0.646 | 0.144 | 0.009 | 0.764 | 0.182 | 0.496 |
| RoadRunner | 0.821 | 0.964 | 0.612 | 0.816 | 0.034 | 0.816 |
| JamesBond | 0.098 | 0.096 | 0.038 | 0.612 | 0.865 | 0.910 |
| CrazyClimber | 0.750 | 0.002 | 0.028 | 0.980 | 0.925 | 0.983 |
| TimePilot | 0.776 | 0.394 | 0.380 | 0.517 | 0.202 | 0.312 |
| Pong | 0.995 | 0.997 | 1.0 | 1.0 | 0.983 | 0.946 |

It is extremely surprising to notice that the impact of $\mathcal{A}^{individual}$ in JamesBond is distinctily lower than $\mathcal{A}_e^{random}$. The reason for this in JamesBond is that $\mathcal{A}_e^{random}$ consistently shifts all actions towards action 12 while $\mathcal{A}^{individual}$ causes the agent to choose different suboptimal actions in every state. See section 4.1 for more details. We observe that this consistent shift towards one particular action results in a much more successful attack in certain environments. In JamesBond, there are certain obstacles that the agent must jump over in order to avoid death, and consistently taking action 12 prevents the agent from jumping far enough. In CrazyClimber, the consistent shift towards one action results in the agent getting stuck in one state where choosing any other action would likely free it. See our github page[1] for videos of the agent's behavior for these cases.

Table 3: Gaussian noise impact for the same $\ell_2$-norm bound used in Table 1 and Table 2.

| | BankHeist | RoadRunner | JamesBond | CrazyClimber | TimePilot | Pong |
|---|---|---|---|---|---|---|
| Impacts | 0.078 | 0.027 | 0.038 | 0.054 | 0.031 | 0.045 |

In all the experiments we set the $\ell_2$-norm bound $\kappa$ to a level so that Gaussian noise with $\ell_2$-norm $\kappa$ has insignificant impact. In Table 3 we show the Gaussian noise impacts on the environments of interest with the same $\ell_2$-norm bound $\kappa$ used in the EAD and Carlini & Wagner (2017) formulations. It is important to note that unbounded Carlini & Wagner (2017) produces perturbations large enough in $\ell_2$-norm that Gaussian noise of the same magnitude has similar impact. Therefore, we bound the perturbation to a level where Gaussian noise is not effective.

## 4.1 EMPIRICAL EVALUATION OF ACTIONS

In this subsection we will define the functions on actions taken by the agent that we will use to understand the effects of the adversarial perturbations. In the following subsections we will use these functions to explore the decision boundaries and representations learned by the deep reinforcement learning agent.

In the OpenAI Atari baselines the discrete action set is numbered from $0$ to $|\mathbb{A}|$. For this purpose for each episode we collected statistics on the probability $P(a)$ of taking action $a$ in the following scenarios:

- $P_{\text{base}}(a)$ - the fraction of states in which action $a \in \mathbb{A}$ is taken by the agent in an episode with no adversarial perturbation added.

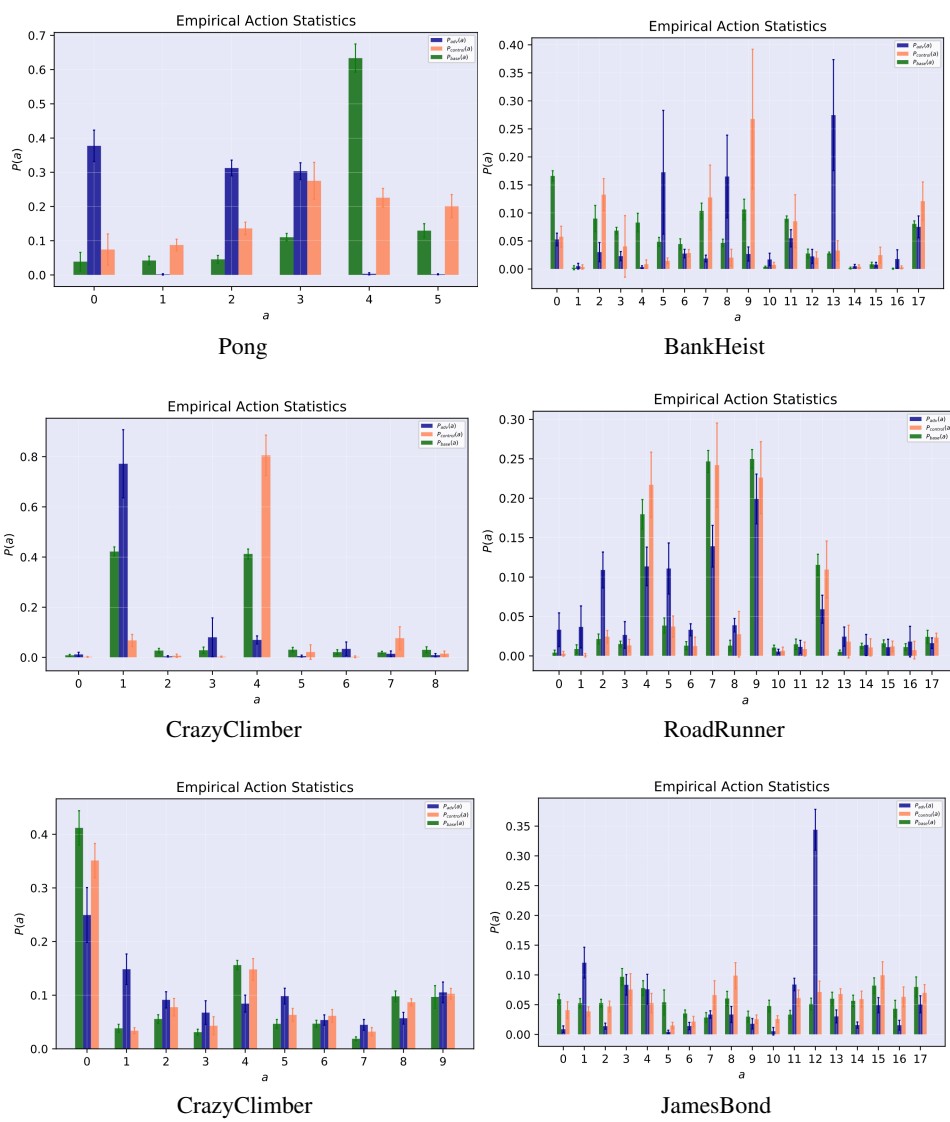

Figure 1: Empirical action statistics of $\mathcal{A}_{\text{e}}^{\text{random}}$ with EAD formulation for $P_{\text{control}}(a)$ , $P_{\text{base}}(a)$ and $P_{\text{adv}}(a)$.

---

[1]  https://adversarialtransferabilitydeeprl.github.io

- $P_{\text{adv}}(a)$ - the fraction of states in which action $a \in \mathbb{A}$ is taken by the agent in an episode with the adversarial perturbation added.

- $P_{\text{adv}}(a, b)$ - the fraction of states in which action $a \in \mathbb{A}$ *would have been* taken by the agent if there were no perturbation, but action $b \in \mathbb{A}$ was taken due to the added perturbation.

- $P_{\text{control}}(a) = \sum_b P_{\text{adv}}(a, b)$ - the fraction of states in which action $a \in \mathbb{A}$ *would have been* taken by the agent if there were no perturbation, in an episode with the adversarial perturbation added.

The difference between $P_{\text{control}}(a)$ and $P_{\text{adv}}(a)$ measures how effectively the adversary changes the action taken by the agent in each state. We refer to this as *action manipulation*. As a baseline an adversary must conduct action manipulation in at least one state. The difference between $P_{\text{control}}(a)$ and $P_{\text{base}}(a)$ measures how effectively the adversary changes the distribution of states visited by the agent. We refer to this as *state manipulation*. Unlike action manipulation it is not readily apparent that significant state manipulation is necessary to achieve a high impact on the agent's performance.

In Figure 1 we observe that in Roadrunner, JamesBond, and TimePilot $P_{\text{control}}(a)$ and $P_{\text{base}}(a)$ are similar. Thus, for these environments state manipulation is relatively small. However, significant action manipulation is occurring. In Pong, BankHeist, and CrazyClimber the adversary achieves both significant state manipulation and action manipulation.

## 4.2 ACTION MANIPULATION

In Figure 2 we show the heatmaps corresponding to $P_{\text{adv}}(a, b)$. In these heatmaps we show which particular control action has been shifted to a particular adversarial action. In some environments there is a dominant action shift towards one particular adversarial action from one control action as in CrazyClimber or in TimePilot. In some games there are one or more clear bright columns where all the control actions are shifted towards the same particular adversarial action, for instance in JamesBond. Recall that all experiments were conducted using a single adversarial offset vector added to every visited state. Thus, the presence of a bright column indicates that, for many different input states, this vector points across a decision boundary that separates one particular action from

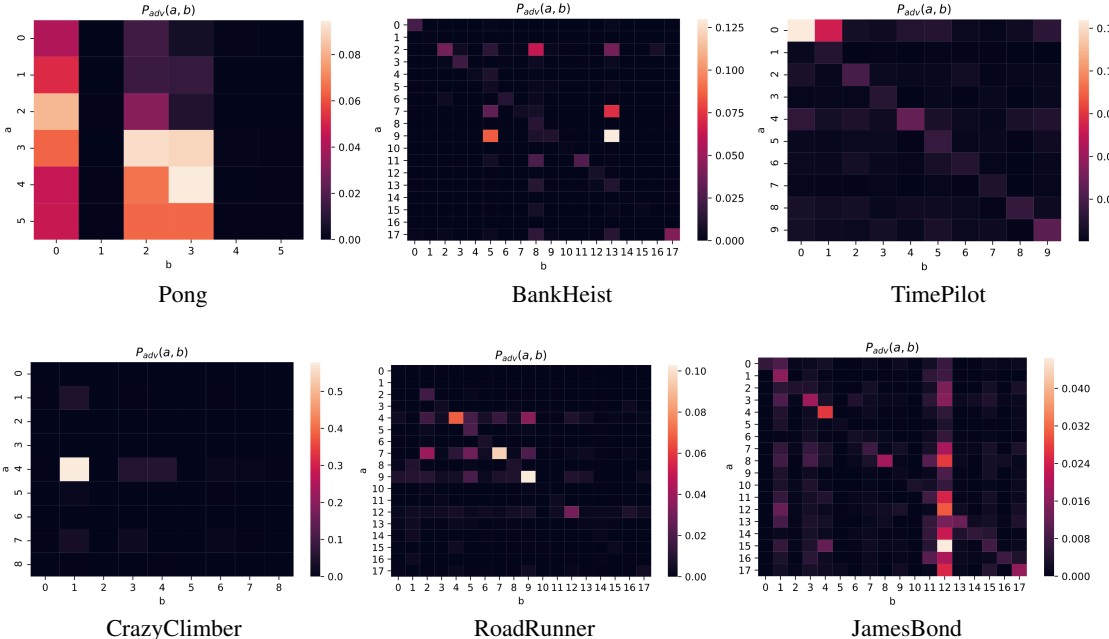

Figure 2: Heat map of $P_{\text{adv}}(a, b)$ for $\mathcal{A}_{\text{e}}^{\text{random}}$ with EAD formulation.

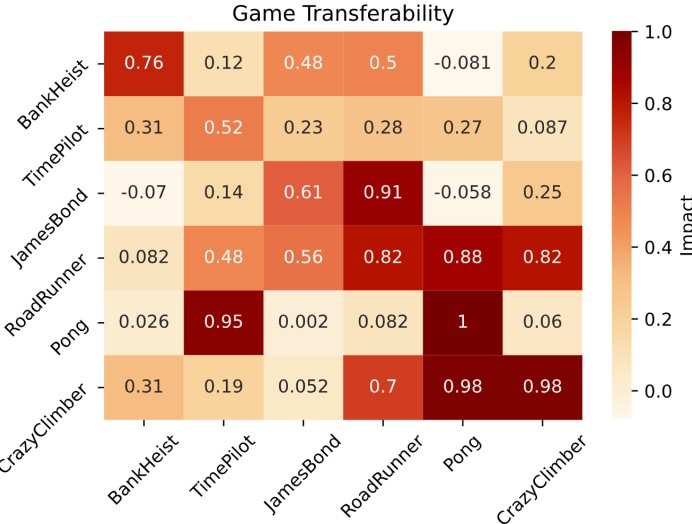

Figure 3: Environment transferability for $\mathcal{A}_{\mathrm{M}}^{\mathrm{random}}$ with EAD formulation. Each row shows the environment where the perturbation is added, and each column shows the environment from which the perturbation is computed.

all other actions. A plausible explanation for these bright columns is that each one corresponds to a diverse set of states for which the neighboring decision boundary geometry is highly correlated.

### 4.3 ENVIRONMENT TRANSFERABILITY

In this section we investigate transferability between environments. In these experiments we utilize $\mathcal{A}_{\mathrm{M}}^{\mathrm{random}}$ where the perturbation is computed from a random state of a random episode of one environment, and then added to a completely different environment. In Figure 3 we show the impacts for the $\mathcal{A}_{\mathrm{M}}^{\mathrm{random}}$ adversary in six different environments. Each row shows the environment where the perturbation is added, and each column shows the environment from which the perturbation is computed. Note that the diagonal of the environment transferability matrix represents state transferability, while the off diagonal elements represent environment transferabilty. One can observe intriguing properties of the Atari baselines where certain environments are highly transferable. We show that the perturbations computed from RoadRunner are highly effective in other environments and the perturbations computed from other environments are effective in RoadRunner. Additionally, in half of the environments higher or similar impact is achieved when transferring a perturbation from a different environment rather than using one computed in the environment itself. This high level of environment transferability is a sign deep reinforcement learning agents are learning representations that have correlated features across different environments.

## 5 CONCLUSION

In this paper we introduce a framework based on six different adversaries within our proposed transferability threat model. Using our framework we investigate the transferability properties of the environments and algorithms used for deep reinforcement learning training. We show that using a perturbation computed from a random state of a random episode of one environment can cause significant detrimental effects in a completely different environment. While this transferability brings several complications from the security point of view, where an attacker can manipulate an agent in a black-box setting without having access to the training environment and neural architecture, we also argue that it is a clear sign that deep reinforcement learning agents indeed learn representations that generalize across different environments. In particular, environment transferability of perturbations indicates that agents have promising underlying generalization skills. We believe our results can give us a perspective on the generalization capabilities of deep reinforcement learning agents, and can be instrumental in building reliable and robust deep reinforcement learning agents.

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

## A  APPENDIX

### A.1  INVESTIGATION OF POLICY GRADIENT METHODS

In this section we provide an investigation into policy gradient algorithms. In Table 4 we demonstrate the impact results of the adversary definitions within the proposed adversarial framework with the EAD formulation. Note that the $\mathcal{A}_e^{\text{random}}$, $\mathcal{A}_e^{\text{initial}}$, $\mathcal{A}_M^{\text{initial}}$, and $\mathcal{A}_M^{\text{random}}$ adversaries in this example are computed by using one single perturbation from a random state of an agent trained with DDQN and then applying this perturbation to the agent trained with A3C. It is quite striking to observe that compared to Gaussian noise the $\mathcal{A}_M^{\text{random}}$ adversary has great success on degrading the agent's performance. Recall that the $\mathcal{A}_M^{\text{random}}$ adversary is computing a single adversarial perturbation from a random state of a completely different MDP trained with a different algorithm.

Table 4: Impacts of the adversary definitions within the proposed adversarial framework as applied to A3C trained agent's observation system where the perturbation is computed with the EAD formulation for an agent trained with DDQN.

| Environment | $\mathcal{A}^{\text{Gaussian}}$ | $\mathcal{A}_e^{\text{initial}}$ | $\mathcal{A}_e^{\text{random}}$ | $\mathcal{A}_M^{\text{initial}}$ | $\mathcal{A}_M^{\text{random}}$ |
|---|---|---|---|---|---|
| Pong | 0.0789 | 0.783 | 0.751 | 0.724 | 0.632 |

## A.2 PERCEPTUAL SIMILARITIES

In this section we provide an analysis of the perceptual similarities between states in the same MDP and states from different MDPs. In particular the perceptual similarity distance $\mathcal{P}$ measures the difference between two inputs based on internal activations of a neural network independent from the network architecture Zhang et al. (2018). Figure 4 demonstrates the perceptual similarity distance between states of different MDPs and states in the same MDP. Note that the perceptual similarity between states is not correlated with environment transferability as shown in Figure 3. In particular, the transferability from Pong to CrazyClimber is high as can be seen in Figure 3, while the perceptual similarity distance between states from the two MDPs is also high (i.e the states are perceptually dissimilar). Thus the perturbations transfer between these two MDPs despite the perceptual dissimilarity between the states of the two different MDPs.

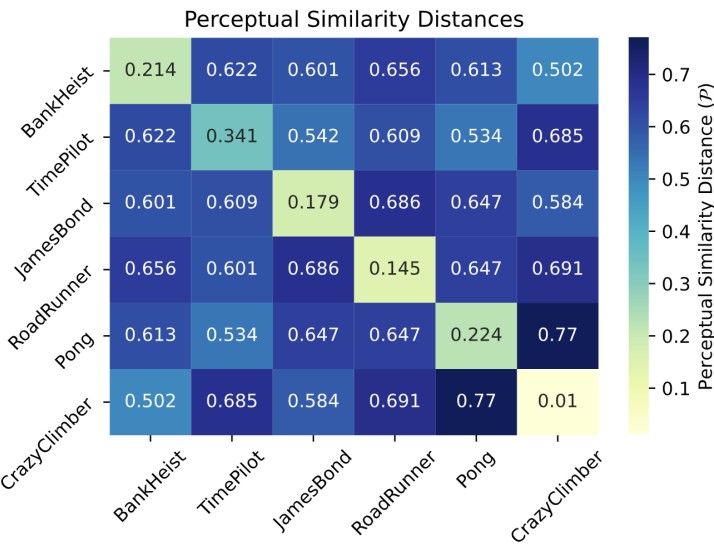

Figure 4: Perceptual similarity distances between states inside the same MDP and between different MDPs.

## A.3 ARCHITECTURAL DIFFERENCES AND HYPERPARAMETERS

In this section we provide transferability results between different architectures for $\mathcal{A}_{\mathrm{e}}^{\mathrm{random}}$ and $\mathcal{A}_{\mathrm{M}}^{\mathrm{random}}$ adversaries. In general we found no difference between DDQN architectures. In the experiments in this section the perturbation is computed in the DDQN Prior Duel architecture, and then added to the agent's observation system trained with DDQN, DDQN Prior, DDQN Duel and DDQN Prior Duel.

Table 5: Impacts of the adversary definitions within the proposed adversarisal framework where the perturbation is computed in the DDQN Prior Duel architecture and added to the agent's observation system trained with DDQN, DDQN Prior, DDQN Duel and DDQN Prior Duel for the Pong environment.

| Architectures | $\mathcal{A}^{\mathrm{Gaussian}}$ | $\mathcal{A}_{\mathrm{e}}^{\mathrm{initial}}$ | $\mathcal{A}_{\mathrm{e}}^{\mathrm{random}}$ | $\mathcal{A}_{\mathrm{M}}^{\mathrm{initial}}$ | $\mathcal{A}_{\mathrm{M}}^{\mathrm{random}}$ |
|---|---|---|---|---|---|
| DDQN | 0.002 | 1.0 | 1.0 | 1.0 | 0.995 |
| DDQN Prior | 0.075 | 1.0 | 1.0 | 1.0 | 0.985 |
| DDQN Duel | 0.036 | 1.0 | 1.0 | 1.0 | 0.966 |
| DDQN Prior-Duel | 0.041 | 1.0 | 1.0 | 1.0 | 0.981 |

DDQN hyperparameters as follows: learning rate for Adam optimizer is $5 \times 10^{-5}$, buffer size is $50000$, final value of random action probability $0.02$, batch size $32$, discount factor is $1.0$. See more details on implementation and hyperparameters in Dhariwal et al. (2017).

## A.4 ENVIRONMENTS

In Figure 5 we show the screenshots of the MDPs considered in this paper. We observe that some environments from the Atari baselines have extremely high transferability. For the extreme example, the impact of the EAD formulation in Pong is $0.974$ for $\mathcal{A}_e^{\text{random}}$. Moreover, it is $0.906$ for $\mathcal{A}_M^{\text{random}}$ Note that this means the perturbation is computed based on a random state of a random episode from a random environment $M$, and still causes the agent to perform extremely poorly.

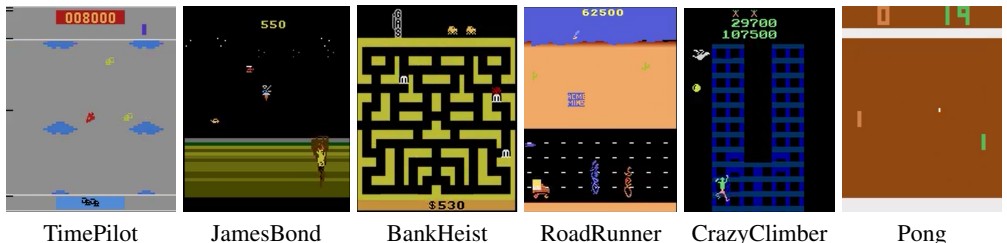

TimePilot    JamesBond    BankHeist    RoadRunner    CrazyClimber    Pong

Figure 5: Different environments from Atari Baselines

