# OpenReview forum: "Exploring Transferability of Perturbations in Deep Reinforcement Learning"
_ICLR.cc/2021/Conference — Reject_

### Official Review · AnonReviewer2 · 2020-10-25
**Needs another round of editing pass**

**Rating:** 4
**Confidence:** 4

**Review:**

— idea:
A framework composed of 6 different adversaries is proposed using which it is claimed that the transferability properties among Atari environments can be studied.

— comments:

The paper is well-written and easy to follow. However, I have major concerns about the novelty of the material discussed in the paper. The only novelty is about using which states as adversarially perturbed input. It mostly looks like an incremental study.

In the experiments section, authors have used a pre-trained DDQN agent which originally does not show significant generalizability compared to methods like A3C. It would be interesting to see the impact of the proposed framework on algorithms with better generalization capabilities. Thus, I believe the experiment section lacks this very important part.

In section 4.2, it is discussed which actions are switched to which ones due to adversary. For example, in the case of Pong, all of the actions are changed to 0, 2, or 3, which are “no op”, “right”, and “left”, respectively. I speculate that a better generalized policy (like A3C) instead of DDQN could still perform well since it has all the actions required to get the maximum return. Thus, the impact would not be this high (according to tables 1 and 2). In addition, in Pong, agent needs to press the “fire” button (action 1) to start each round of the game. Without loss of generality, this could be overwritten in an engineering matter (like what has been done in DeepMind Atari wrapper) so the agent still be able to perform without being concerned about the “fire” button. There is a lot to investigate and talk about in this paper.

— minor issues:
In figure 1, there are two CrazyClimber plots. One of them needs to be changed.

I believe the paper in its current form requires substantial changes. Due to the aforementioned reasons, I don't feel comfortable supporting this version of the paper. I think this paper could benefit from another round of editing pass.

---

> ### Author Response · Authors · 2020-11-15
> **Author Response**
>
> Thank you for your comments and suggestions.
>
> 1. "The only novelty is about using which states as adversarially perturbed input.":
>
> We believe that the main novelty in the paper is our exploration of the fact that a single, fixed adversarial perturbation computed from a random state of one MDP can be used to cause an agent to fail in a completely different MDP. Prior work on adversarial examples in DRL has focused on computing new perturbations customized to either each state of the game (Huang et al. (2017)) or customized to the set of states chosen to be attacked (Sun et al. (2020); Lin et al. (2017)). This requires the ability to access and modify the state observations of the agent in real-time before the agent perceives the states, whereas the $A_M^{random}$ adversary in our paper does not. In particular, our paper shows that by computing a single perturbation from a random state of a completely different MDP it is possible to substantially degrade the performance of trained DRL agents. This was not known before and we find it quite surprising.
>
> 2. "I speculate that a better generalized policy (like A3C) instead of DDQN could still perform well since it has all the actions required to get the maximum return."
>
> At your suggestion we tested an A3C agent in Pong using our framework. We computed a single perturbation from a completely different MDP TimePilot trained with DDQN, and added that single perturbation to the states visited by the A3C agent in Pong. This still results in large impact, showing that perturbations computed in one MDP with a different algorithm do indeed transfer to A3C agents. Please see Appendix Section A.1.
>
> 3. " In figure 1, there are two CrazyClimber plots. One of them needs to be changed.":
>
> Thanks for catching this. We have fixed this in the paper.
>
> 4. “In addition, in Pong, agent needs to press the “fire” button (action 1) to start each round of the game. Without loss of generality, this could be overwritten in an engineering matter (like what has been done in DeepMind Atari wrapper) so the agent still be able to perform without being concerned about the “fire” button.”
>
> As we cite in the paper we use OpenAI Gym Brockman et. al (2016) not DeepMind Atari wrapper. OpenAI Gym Pong action set size is 6 with the following meanings [‘NOOP’, ‘FIRE’, ‘RIGHT’, ‘LEFT’, ‘RIGHTFIRE’, ‘LEFTFIRE’].The action size 6 is redundant where ‘FIRE’ is equivalent to ‘NOOP’, ‘LEFT’ is equivalent to ‘LEFTFIRE’, and ‘RIGHT’ is equal to ‘RIGHTFIRE’. In particular, for the numbers used in Figure 1 and Figure 2 action ‘0’ is equivalent to action ‘1’, action ‘2’ is equivalent to action ‘4’ and action ‘3’ is equivalent to action ‘5’. We could have changed this order and reported the action set consisting of 3 actions. However, we did not want to change the official definition of the OpenAI baseline where it might lead to confusion and might hurt reproducibility. That is why we cited the OpenAI Gym Brockman et. al (2016) and used their definition of the action set.
>
> If we did understand your concern about the action ‘FIRE’ correctly the agent is indeed taking the action ‘1’ for $P_{adv}, P_{base}, $, and $P_{control}$, because action ‘1’ is equivalent to action ‘0’. This also can be seen in Figure 1. Please let us know if you have more questions on this matter. We would be glad to address them.
>
> Moreover, please see Figure 2 for the heatmap of action changes in Pong. The Figure 2 heatmap shows the maximum action shift for action ‘0’ is towards action ‘2’, and the maximum action shift for action ‘2’ is towards action ‘0’. Note that action ‘0’ is equivalent to action ‘1’. Thereby, shifting from action ‘2’ to action ‘0’ will indeed have an effect on agent’s performance. Again similarly the maximum shift for action ‘4’ is towards action ‘3’ and note that action ‘4’ is equivalent to action ‘2’. Thereby, shifting from action ‘4’ to action ‘3’ will indeed have an effect on the agent’s performance.
>
> Again please let us know if you would have any further questions on this. We would be glad to address them.
>
> Greg Brockman, Vicki Cheung, Ludwig Pettersson, Jonas Schneider, John Schulman, Jie Tang, and Wojciech Zaremba. Openai gym. arXiv:1606.01540, 2016.

---

> > ### Author Response · Authors · 2020-11-22
> > **Author Response Comments**
> >
> > We hope that our response addressed your comments and questions. We added a section in Appendix A.1 on your suggestion regarding policy gradient methods. Please let us know if you have any further comments or questions.

---

> ### Author Response · Authors · 2020-11-25
> **Discussion Period**
>
> We didn't receive any feedback on our author response in the discussion period. We hope your questions and comments have been addressed. We appreciate your time and effort in your initial review. We hope that you can take our author response into consideration in your review update.

---

### Official Review · AnonReviewer4 · 2020-10-26
**Interesting title but bad approach**

**Rating:** 3
**Confidence:** 4

**Review:**

Paper Summary:
This paper aims at discovering transferability of perturbations across different environments in RL. The authors propose some different types of advasaries and tested those adversaries on 6 atari games.

Review Summary:
The idea of analyzing perturbations' transferability is interesting, but it is hard to say that the approaches are satisfactory. The authors did not propose any novel algorithms or modifications based on existing algorithms, and provide only preliminary experiment results. As such, I suggest a clear rejection.

Detailed Comments:
Section 3.1, paragraph 2. "Such a model... of the deep reinforcement learning agent". I am hardly convinced by this statement and led by this statement, the proposed approach in this work. When we consider transferability in machine learning, we assume that there are something common to learn between the two domains. My first thought when I read the title is that it would be a transfer learning method for adversarial attacks. But the authors seem to prefer a universal offset on the raw pixel input. In this case, the only insights we might possibly learn is the input similarity between different Atari games, which is hardly a contribution to any field. Moreover, since adversarial attack is already used in this work, I don't see the reason why none of the referred methods for adversarial attacks in RL are tested.

Besides, all experiments are done only on DDQN, hence the claims are hardly validated. What I expect is either a new algorithm that is more resilient or a comparison on robustness between different RL algorithms. It is also not explained how the environments are chosen.

---

> ### Author Response · Authors · 2020-11-15
> **Author Response Part 2**
>
> 4.  “the only insights we might possibly learn is the input similarity between different Atari games”:
>
> This is not true. The reason is that similarity between inputs alone is not sufficient for transferability of perturbations, because the effectiveness of the perturbation depends on both the input and the entire trained model. Please see more on transferability of adversarial examples in Szegedy et al. (2014), Goodfellow et al (2015), Tramer et al (2017), Liu et al. (2017), Dezfooli et al. (2017), Dong et al (2018), Tramer et al. (2017), and Dongxian et al. (2020).
>
> Even if the inputs were quite similar, Deep RL models trained for different games perform semantically quite different tasks, and so it is not a priori obvious that a perturbation effective in one game will transfer to another. Indeed there are some pairs of games where perturbations do not transfer well, and some where they do. Further, this does not seem correlated with the perceptual similarity of the inputs for the pairs of games as shown in Appendix A.2.
>
> 5. "Besides, all experiments are done only on DDQN, hence the claims are hardly validated":
>
> We also added a new section dedicated to policy gradient evaluation in the Appendix Section A.1. Please let us know if you have any further questions. We would gladly address them.
>
> 6.  Would it be possible to elaborate on “ why none of the referred methods for adversarial attacks in RL are tested”? We were a little bit confused by what was meant by this comment.
>
>  7.  "It is also not explained how the environments are chosen.":
>
> Environments are chosen based on the diversity in their action set sizes,  the perceptual dissimilarity of the states, and the semantic differences between the tasks learned in different games from the Atari Baselines. In particular, for perceptual similarity differences see Appendix Section A.2. For action size  and semantic differences please see Bellemare et. al. (2013). For a small example BankHeist ([BankHeist-v0](https://gym.openai.com/envs/BankHeist-v0/)) and Amidar ([Amidar-v0](https://gym.openai.com/envs/Amidar-v0/)) can be categorized as relatively more similar based on the semantics of the learned tasks and the perceptual similarity of the states. Also note that DRL algorithms still cannot perform well across all the games uniformly (Wang et. al. (2016)). Therefore, we also did not focus on the games where the algorithm does not perform well yet. Also please see related work on adversarial deep reinforcement learning on the set of games used in Lin et. al (2017), Sun et. al. (2020), Huang et. al. (2017).
>
> Marc G Bellemare, Yavar Naddaf, Joel Veness, and Michael. Bowling. The arcade learning environment: An evaluation platform for general agents. Journal of Artificial Intelligence Research., pp. 253–279, 2013.
>
> Ziyu Wang, Tom Schaul, Matteo Hessel, Hado Van Hasselt, Marc Lanctot, and Nando. De Freitas. Dueling network architectures for deep reinforcement learning. Internation Conference on Machine Learning ICML., pp. 1995–2003, 2016.

---

> > ### Author Response · Authors · 2020-11-22
> > **Author Response Comments**
> >
> > We hope that our response addressed your comments. Please let us know if you have any further comments or questions that you would like to have addressed.

---

> ### Author Response · Authors · 2020-11-15
> **Author Response Part 1**
>
> 1. “When we consider transferability in machine learning, we assume that there are something common to learn between the two domains. My first thought when I read the title is that it would be a transfer learning method for adversarial attacks.”:
>
> The transferability of adversarial examples is a well studied field. See the transferability of adversarial examples in Szegedy et. al. (2014), Goodfellow et al. (2015), Tramer et al. (2017), Liu et al. (2017), Dezfooli et al. (2017), Dong et al. (2018), Tramer et al. (2017), and Dongxian et al. (2020).
>
> [Dongxian et al. (2020)] Dongxian Wu, Yisen Wang, Shu-Tao Xia, James Bailey, Xingjun Ma: Skip Connections Matter: On the Transferability of Adversarial Examples Generated with ResNets. ICLR 2020.
>
> [Goodfellow et al. (2015)] Ian J. Goodfellow, Jonathon Shlens, Christian Szegedy: Explaining and Harnessing Adversarial Examples. ICLR (Poster) 2015.
>
> [Szegedy et al. (2014)] Christian Szegedy, Wojciech Zaremba, Ilya Sutskever, Joan Bruna, Dumitru Erhan, Ian J. Goodfellow, Rob Fergus: Intriguing properties of neural networks. ICLR (Poster) 2014.
>
> [Liu et al. (2017)] Yanpei Liu, Xinyun Chen, Chang Liu, Dawn Song: Delving into Transferable Adversarial Examples and Black-box Attacks. ICLR (Poster) 2017
>
> [Tramer et al (2017)] Florian Tramèr, Nicolas Papernot, Ian J. Goodfellow, Dan Boneh, Patrick D. McDaniel: The Space of Transferable Adversarial Examples. CoRR abs/1704.03453 (2017)
>
> [Dong et al. (2018)] Yinpeng Dong, Fangzhou Liao, Tianyu Pang, Hang Su, Jun Zhu, Xiaolin Hu, Jianguo Li: Boosting Adversarial Attacks With Momentum. CVPR 2018: 9185-9193.
>
> [Dezfooli et al. (2017)] Seyed-Mohsen Moosavi-Dezfooli, Alhussein Fawzi, Omar Fawzi, Pascal Frossard: Universal Adversarial Perturbations. CVPR 2017: 86-94.
>
> 2. "Such a model... of the deep reinforcement learning agent". I am hardly convinced by this statement and led by this statement, the proposed approach in this work.”:
>
> The importance of black-box attacks is also a quite well studied field. Please see for more information on this Papernot (2017), Eykholt et al. (2018), liu et al. (2017), Dong et al. (2018), Li et al. (2019), Athalye et al. (2018).
>
> In particular the Carlini-Wagner adversarial formulation takes 10000 iteration to lower the $\ell_2$-norm of the adversarial perturbation to 0.1 and one might agree that this is even more computationally expensive when the perturbation is computed custom to each state and added to agent’s observation in real time before the agent perceives the state. In comparison our adversarial framework from Definition 2 through Definition 6 only requires one fixed pre-computed adversarial offset vector, resulting in a computation which is significantly more efficient, and which also does not have to be performed in real time as the agent moves to new states.
>
> We also did not expect it to be controversial to claim that having the ability to apply a new perturbation to each of the agent’s observations in real-time is a very strong assumption. It seems reasonable that one might not expect this assumption to hold in many real-world applications e.g. the perception system of a self driving car. In particular, assuming that one has  the ability to both access and modify a self driving car’s perception system in real time, before the state is observed by the car, is a stronger assumption than simply having the ability to apply a sticker to the car’s camera lens. Since the approach in our paper relies on only a single fixed adversarial offset vector, it could also plausibly be implemented by camera stickers as in (Li et. al. 2019).
>
> [Athalye et al. (2018)] Anish Athalye, Logan Engstrom, Andrew Ilyas, Kevin Kwok: Synthesizing Robust Adversarial Examples. ICML 2018: 284-293.
>
> [Eykholt et al. (2018)] Kevin Eykholt, Ivan Evtimov, Earlence Fernandes, Bo Li, Amir Rahmati, Chaowei Xiao, Atul Prakash, Tadayoshi Kohno, Dawn Song: Robust Physical-World Attacks on Deep Learning Visual Classification. CVPR 2018: 1625-1634.
>
> [Li et al. (2019)] Juncheng Li, Frank R. Schmidt, J. Zico Kolter: Adversarial camera stickers: A physical camera-based attack on deep learning systems. ICML 2019
>
> 3. “The authors did not propose any novel algorithms or modifications based on existing algorithms":
>
> We do not claim to introduce a new algorithm. We propose an adversarial framework to investigate the environment and state transferability properties of the perturbations on trained DRL agents defined in Section 3.0. Our framework lays out intriguing unknown properties of the environments used in DRL training. Our paper shows that a single adversarial perturbation computed in one MDP can have large impact in an entirely different MDP. This was not known before and we find it quite surprising.

---

> ### Author Response · Authors · 2020-11-25
> **Discussion Period**
>
> We didn't receive any feedback on our author response in the discussion period. We hope your comments have been addressed. We appreciate your time and effort in your initial review. We hope that you can take our author response into consideration in your review update.

---

### Official Review · AnonReviewer1 · 2020-10-28

**Rating:** 6
**Confidence:** 3

**Review:**

### Summary of Contributions

The paper explores adversarial perturbations in deep RL, providing a new thread model where the perturbation is computed based on a single state. The paper explores the impact of various types of perturbations between states in the same environment (state transferability), and between states in different environments (environment transferability).

### Review

I think the paper does a good job at capturing the extent/prevalence of the issue of adversarial perturbations. I liked the breadth in the types of perturbations considered, and how they map to scnarios that could happen in practice (e.g., restricted adversaries, etc.).

One thing that I think could improve the paper is teasing apart properties of environments or learning algorithms that are telling of the transferrability of the perturbations. For example, beyond suggesting that the offset pushes things beyond the decision boundary, checking things like how action gaps change may be insightful. Some of the action manipulation may further be exacerbated by things like using ε-greedy behavior policies which immediately snaps to highest-valued actions, in contrast with things like Boltzmann policies over action-values or policy gradient methods, which are smooth with respect to changes in them. Do the authors have any insight as to how the trends might change should a smooth policy be used?

While the focus is on deep RL, perhaps a simple, motivating example (e.g., a little gridworld or Markov chain) could make a stronger/clearer case as to what exactly is happening, and suggest what situations one might expect it to be a more prevalent issue.

Beyond this, I have the following questions/concerns:

1) Is there a reason for the choice of DDQN, over say, regular DQN? While one algorithm may perform a bit better, if the emphasis of the paper is to measure the extent of adversarial perturbations, it seems like it may paint a clearer/more convincing picture if a simpler algorithm is used, with fewer moving parts to attribute performance differences to. Along these lines, I think it would be more convincing to carry out the same experiments with another deep RL algorithm (perhaps a policy gradient one, to be representative of both value-based and policy-based methods) to see if comparable levels of and trends in transferrability are observed there. Can the authors comment on whether

2) How were hyper-parameters chosen for the DDQN agent, and can the authors comment on whether such choices can have reasonably strong interplay with the transferability of perturbations? For example, would larger/smaller learning rates, or deeper vs wider networks be more or less resilient to such perturbations? I think in any case such details need to be included in the paper, to be clearer that the transferability quantified is in the scope of a particular instantiation of a specific algorithm.

Overall, I'm erring toward acceptance in that it outlines an interesting framework to study adversarial attacks in deep RL, and provides some early empirical results and intuitions around them. I do think the paper falls a little short in that there wasn't a representative sample of deep RL methods, as well as not commenting on design choices made and how/whether they might interplay with the transferability measured. I'm willing to adjust my score should my concerns be addressed.

---

> ### Author Response · Authors · 2020-11-15
> **Author Response**
>
> Thanks for your insightful comments and useful suggestions! We tried to address them below.
>
> 1.	It is a great suggestion to recommend that we investigate policy gradients. We added a section in the Appendix Section A.1 dedicated to investigation of policy gradient methods. Please let us know if you have any  further questions.
>
> 2.	The reason we focused on DDQN architectures is that they are known to be well established and to perform well. Our concern with DQN was that demonstrating a weakness in an algorithm that is known to perform worse could be perhaps seen as a weakness of the adversary. However, as you suggested we added hyperparameters, and results on investigation of different architectures in Appendix Section A.3.
>
> Please let us know if you have any further questions. We would be glad to address them.

---

> > ### Author Response · Authors · 2020-11-22
> > **Author Response Comments**
> >
> > We hope that our response addressed your questions and comments. We added two new sections in the Appendix A.1 and A.2 based on your suggestions. Thank you again for your useful suggestions and detailed feedback. Please let us know if you have any other comments or suggestions that you would like to have addressed.

---

### Official Review · AnonReviewer3 · 2020-11-01
**Work about stabilities in deep reinforcement learning, with space for improvement**

**Rating:** 4
**Confidence:** 4

**Review:**

The authors studied how perturbations on states would affect the performance of deep reinforcement learning. They defined different types of perturbations, like different perturbations for each state, or apply the perturbations on the initial state on every state. The authors tested these perturbations in some existing environments.
- The conclusions of this submission are unclear and questionable. The authors showed many results in their submission, but all the conclusions are plausible. The results look pretty random, and I do not believe we can draw conclusions from them. For example, in table 2, the impacts of A_M^random are large, even comparable with A^individual. The A_M^random is more like a fixed random noise, so we cannot draw conclusions about transferability from it. It would be helpful to add two new baselines, one is fixed random noise, another is iid random noise, to demonstrate these adversaries are different from random noise.
- Equations 1 and 2 do not make sense. For the equations wrote by the authors, J(s) is a constant, and we should always let s_adv equal to s.

--Post Rebuttal--
Thank the authors for the response. I agree with other reviewers that the task is interesting and the submission has great potential, but might need another round of editing. The results show "a single perturbation from a random state of a completely different MDP" is not normally distributed. However, I agree with R4 that applying a single perturbation from a random state of another MDP does not make sense unless the two MDP and the two states have some similarities.

---

> ### Author Response · Authors · 2020-11-15
> **Author Response**
>
> We have tried to address your comments below. I hope we were able to clear up any confusion. Please let us know if you have any more questions. We would gladly try to address them.
>
> 1. “It would be helpful to add two new baselines, one is fixed random noise” :
>
> We do already have this information in the paper in Table 3 which shows impacts of fixed random Gaussian noise with $\ell_2$-norm equal to the perturbation types used in Table 1 and Table 2.
>
> 2. ” The $A_M^{random}$ is more like a fixed random noise,”:
>
> Item (1) above shows that $A_M^{random}$ is not a fixed random noise as it has much higher impact compared to Gaussian noise with the same magnitude.
>
> 3. “in table 2, the impacts of $A_M^{random}$ are large, even comparable with $A^{individual}$”:
>
> Yes exactly. As we mentioned above in item (2) this impact greatly outperforms random noise and indeed as you point out is comparable to adversarial perturbations computed for each individual state of the game. This shows that by computing a fixed perturbation in one MDP we can have a large impact in another MDP i.e. these fixed perturbations transfer in a black-box setting for environments and states.
>
> 4. "The results look pretty random, and I do not believe we can draw conclusions from them":
>
> Based on above item (1), item(2), item (3) and Table 1 through Table 3 we can indeed draw the conclusion that by computing only a single perturbation from a random state of a completely different MDP an adversary can degrade the performance of a trained DRL agent substantially more than a Gaussian perturbation with the same magnitude. In particular, this is also clear evidence of environment and state transferability as defined in Section 3.0.
>
> 5. “Equations 1 and 2 do not make sense.”
>
> Thank you for pointing out the typos in Equation 1 and 2. We corrected them. Please let us know if you have any more questions. We would gladly try to address them.

---

> > ### Author Response · Authors · 2020-11-22
> > **Author Response Comments**
> >
> > We hope that our response addressed your question. Please let us know if you have any further comments or questions that you would like to have addressed.

---

> ### Author Response · Authors · 2020-11-25
> **Discussion Period**
>
> We didn't receive any feedback on our author response in the discussion period. We hope your question and comments have been addressed. We appreciate your time and effort in your initial review. We hope that you can take our author response into consideration in your review update.

---

### Author Response · Authors · 2020-11-24
**Author Meta Response**

Dear Area Chairs and Reviewers,

We responded to each reviewer individually. Here we summarize the revisions to the paper during the author rebuttal period below:

* **[15.11.2020]** We added a section on policy gradient evaluation in Appendix Section A.1.

* **[15.11.2020]** We added a section dedicated to investigation of the adversarial framework with different DDQN architectures in Appendix section A.3.

* **[15.11.2020]** We added a section dedicated to perceptual similarities between different environments to further clarify the potential underlying reasons for environment transferability in Appendix Section A.2.

* **[15.11.2020]** We added the relevant hyperparameters in Appendix Section A.3.

---

### Decision · Program_Chairs · 2021-01-07
**Final Decision**

**Decision:**

Reject

**Comment:**

The work studies the transferability of perturbations/adversarial attacks on DRL agents. As a way to mitigate the cost of generating individual perturbation for each state in each episode, the authors proposed several variants to use same perturbation across different states across different episodes. While reviewers recognize the potential of the direction, they are not comfortable accepting the paper at its current state. The experiment results in its current form does not provide enough support to the claim. In particular, it is unclear how much the shared perturbation changes the original perception in comparison to the individual comparison, and how should the impact number differences be interpreted. Reviewers brought up concerns regarding all the experiments being evaluated on a DDQN agent, and not enough clarities has been provided on the different design choices. If perceptual similarity is not the indicator of environment transferability, do the authors have intuition on what does?